# Thyroid Hormones in Early Pregnancy and Birth Weight: A Retrospective Study

**DOI:** 10.3390/biomedicines13030542

**Published:** 2025-02-21

**Authors:** Marco La Verde, Pasquale De Franciscis, Rossella Molitierno, Florindo Mario Caniglia, Mario Fordellone, Eleonora Braca, Carla Carbone, Claudia Varro, Paolo Cirillo, Lorenzo Scappaticcio, Giuseppe Bellastella

**Affiliations:** 1Department of Woman, Child and General and Specialized Surgery, Obstetrics and Gynecology Unit, University of Campania “Luigi Vanvitelli”, 80138 Naples, Italy; pasquale.defranciscis@unicampania.it (P.D.F.); rossella.molitierno@unicampania.it (R.M.); florindomario.caniglia@gmail.com (F.M.C.); eleonorabraca9@gmail.com (E.B.); 2Medical Statistics Unit, University of Campania “Luigi Vanvitelli”, 80138 Naples, Italy; mario.fordellone@unicampania.it; 3Unit of Endocrinology and Metabolic Diseases, AOU University of Campania “Luigi Vanvitelli”, 80138 Naples, Italy; carla.carbone@unicampania.it (C.C.); claudia.varro@unicampania.it (C.V.); lorenzo.scappaticcio@unicampania.it (L.S.); giuseppe.bellastella@unicampania.it (G.B.); 4Department of Advanced Medical and Surgical Sciences, University of Campania “Luigi Vanvitelli”, 80138 Naples, Italy; paolo.cirillo10@hotmail.it

**Keywords:** thyroid function, thyroid diseases, pregnancy, birth weight, neonatal, infant

## Abstract

**Background/Objectives:** Various factors influence intrauterine growth and birth weight. We investigated the possible association between first-trimester pregnancy thyroid functions and birth weight. **Methods:** A retrospective observational study was conducted from 31 March 2021 to 30 September 2022. Ninety-eight low-risk pregnant women were evaluated. To obtain a homogeneous population, we included only patients with no maternal or fetal comorbidities. TSH, FT3, and FT4 levels in the first trimester and birth weight were recorded and analyzed. **Results:** The average maternal age was 33.37 years (IQR = 8.67), with a median BMI of 24.98 kg/m^2^ (IQR = 5.66). The average gestational week of pregnancy was 39.40 weeks (IQR = 2.10). Linear regression for birth weight according to TSH effects adjusted by sociodemographic and clinical factors did not show any associations between birth weight and TSH, age, ethnicity, BMI, smoking, or weight gain. Pregnancy duration was associated with birth weight: β = 172.027, *p*-value < 0.001. A negative significant correlation between FT3 and birth weight was found (beta = −118.901 95% CI: −222.942 to −14.859, *p* = 0.026). Age, ethnicity, BMI, smoking, and weight showed non-significant associations with birth weight. The FT3 scatterplot showed a negative correlation of FT3 levels with birth weight. Higher values of FT3 were associated with a low birth weight (R = −0.22, *p* = 0.029). **Conclusions:** Our study, focused on the first trimester, found a negative correlation between FT3 variations and birth weight.

## 1. Introduction

Different factors impact intrauterine fetal growth. During the earlier stages of pregnancy, thyroid function seems to be crucial [1,2]. At this embryonic stage, the maternal thyroid gland supports fetal growth [3,4]. Thyroid hormones operate at the cellular level as an activator of cellular gene expression. [1,2]. They act by interacting with thyroid hormone receptors to support developing tissues, which include the fetal brain, skeleton, and liver [5]. An absence of thyroid hormones early in pregnancy disrupts these processes, leading to long-lasting cognitive impairments and neurodevelopmental disturbances [6]. A deficiency in thyroid hormones results in delayed skeletal maturation [7].

Hypothyroidism reduces metabolic efficiency to support critical developmental processes, whereas hyperthyroidism overstimulates metabolic pathways associated with enhanced oxidative stress, leading to potential cellular damage [8].

The placenta represents a target and a mediator of thyroid hormone action. During the first trimester, maternal dysfunction can potentially interfere with normal fetal development [5]. Besides the direct hormonal influences, thyroid dysfunction may interfere with other endocrine systems that act complementarily on fetal growth, such as the insulin-like growth factor axis. Impaired thyroid function may thus decrease the activity of IGF-1 and further contribute to growth deficits in the fetus [9].

Thyroid dysfunction has been linked to low birth weight and preterm birth [10,11]. The Generation R Study has provided evidence for an association of maternal thyroid dysfunction with birth weight [12]. Hypothyroidism has been related to increased rates of preterm delivery and neonatal respiratory distress syndrome, with implications for long-term neurodevelopmental outcomes in the offspring [13]. Subclinical hypothyroidism has been linked to adverse outcomes [8]. Other studies have evidenced a gestational diabetes or low-birth-weight association with thyroid dysfunction [14]. The thyrotoxic state in the fetus disrupts normal growth processes and heightens the likelihood of neonatal intensive care admissions secondary to respiratory or metabolic difficulties [15,16]. Adverse pregnancy outcomes heighten the need for thyroid screening in high-risk pregnant populations [17]. The scientific literature has considered a single-time-point concentration of thyroid hormones only during the third trimester of pregnancy [18]. Early pregnancy is a time of considerable thyroidal adaptation, most probably influenced by exogenous factors—such as hCG levels—and increased demand for thyroidal hormones for the developing fetus, and the relationships between early TSH levels and fetal growth represent a critical literature gap [19,20,21].

Fluctuations in TSH levels during early pregnancy may have long-term implications for fetal growth [22]. However, the birth weight difference associated with various degrees of thyroid function has not been completely described, and the role of early-pregnancy thyroid hormone levels in the determination of birth weight is incompletely identified [23]. Birth weight indirectly denotes fetal well-being and development as a result of several interactions and reflects neonatal health and fetal outcomes [24,25,26,27]. The placenta is the central player in this process by providing the oxygen and nutrients necessary for growth [28]. Placental impairment can cause small for gestational age (SGA) or macrosomia [28,29]. SGA represents an estimated fetal weight under the 10th percentile [30]. Macrosomia refers to a birth weight above the 90th percentile, usually due to maternal conditions such as excessive weight gain or uncontrolled gestational diabetes [31]. The risks of delivery complications, asphyxia, and low Apgar values appear to be related to macrosomia and SGA [32]. Ultrasound, Doppler velocimetry, and cardiotocography have a significant role in monitoring fetal growth and fetal well-being [33,34]. Birth weight remains a critical datum in prenatal care. Maternal prepregnancy body mass index (BMI), weight gain, thyroid hormone level, and nutritional status are all significant regulators of fetal growth [25,35,36]. Gestational diabetes and hypertension influence birth weight [37,38]. SGA and macrosomia fetuses are associated with significant perinatal and long-term complications. Understanding their etiologies has become critical to preventive and therapeutic strategies. SGA etiologies include maternal and fetal pathologies that generally reduce uteroplacental blood flow, with consequent impairment of oxygen and nutrient delivery to the fetus [39]. Smoking during pregnancy impairs fetal growth through vasoconstriction and by exposing the fetus to carbon monoxide, which diminishes oxygen transport [39]. Other central factors of SGA include placental dysfunction [39].

Macrosomia is related to maternal hyperglycemia, gestational diabetes, and excessive gestational weight gain [40]. Maternal gestational diabetes with an elevated level of glucose stimulates fetal hyperinsulinemia [40]. Insulin is a potent growth factor that stimulates fat deposition and general fetal size [41]. Large-for-gestational-age infants of diabetic mothers usually have disproportions in growth, with larger shoulders and trunk compared to the head [42]. Other factors also contributing significantly to macrosomia, irrespective of glycemic status, include excessive maternal weight gain during pregnancy [40]. The incidence of macrosomic births tends to be higher in pregnancies with a higher prepregnancy BMI and poor gestational weight management [43]. Excessive maternal weight gain via an unhealthy diet and lifestyle factors predisposes to a macrosomic fetus [44]. SGA and macrosomic conditions are strictly related to neonatal health. For example, SGA newborns are at higher risk of hypothermia and developmental delays in the long term [45]. Postnatal growth patterns may further accentuate these outcomes, with rapid catch-up growth predisposing individuals to metabolic syndrome and cardiovascular diseases later in life [45]. Birth trauma also complicates the macrosomia, with a high incidence of shoulder dystocia with subsequent brachial plexus injury [43]. Macrosomic babies have a higher risk of later obesity and type 2 diabetes [46]. Management and prevention in both SGA and macrosomia can be addressed with a multidisciplinary approach including nutritional counselling, weight management, and optimum management of various preexisting maternal conditions, including diabetes and thyroid disorders [47]. Prenatal interventions include treatment of chronic conditions, appropriate nutrition, smoking cessation, and regular monitoring of fetal growth [47]. All these factors can reduce the risks of abnormal birth weight and improve neonatal outcomes [47].

Our objective was to establish the relationship between the levels of maternal thyroid hormones during early pregnancy and birth weight, particularly concerning subtle variations within the normal physiological range. We postulated that even small variations in maternal thyroid hormones, even within the normal range during early pregnancy, may be related to birth weight.

## 2. Materials and Methods

We conducted a retrospective study, at a tertiary care university hospital, the Luigi Vanvitelli University of Campania Hospital in Naples. The Helsinki Declaration, the Committee on Publication Ethics guidelines (http://publicationethics.org, accessed on 18 October 2024), and the RECORD statement of the Strengthening the Reporting of Observational studies in Epidemiology (STROBE) collaborative [48], provided through the EQUATOR network (https://www.equator-network.org, accessed on 18 October 2024) were considered throughout the design, analyses, interpretation, drafting, and revising processes. Data anonymization included removal of any personal data that could identify the patient. The procedure was explained, and the informed consent form was signed by all patients. The study was approved by the Medical Ethics Committee of the Campania 2 (protocol 0024105/i). To avoid population bias, we included only low-risk pregnancies with maternal age from 18 to 45 years and followed by our obstetric ambulatory antenatal care team between 31 March 2021 and 30 September 2022. The following conditions were excluded: overt hypo- or hyperthyroidism, gestational and pregestational diabetes, chronic hypertension, multiple pregnancies, preterm delivery, gestational hypertension, preeclampsia, and eclampsia. In addition, we excluded fetal pathologies such as fetal growth restriction, stillbirth, genetic disorders, and malformations in the fetus. The following data were collected for each pregnancy from 5 weeks of gestation until 13 weeks: (1) sociodemographic characteristics (age, ethnicity, weight, height, BMI, smoking, weight gain); (2) clinical information concerning the current course of pregnancy (week of gestation, obstetric complications during pregnancy); and (3) TSH, free triiodothyronine (FT3), and thyroxine (FT4) levels measured in early pregnancy (during the first trimester of pregnancy). All pregnant women with abnormal thyroid hormone levels were followed up by our endocrinologist until the postpartum period. This follow-up involved regular assessment of hormone levels, therapeutic intervention if necessary, including levothyroxine or anti-thyroid medication, and additional fetal surveillance by ultrasound and Doppler studies to ensure fetal well-being. The mothers were followed up with a postpartum thyroid evaluation for the detection of potential thyroid dysfunction and readjustment of treatments as necessary to transition into postpartum. Continuous variables are reported as either the means and standard deviation or median and interquartile ranges (IQRs) according to their distribution, as assessed by the Shapiro–Wilk normality test. Categorical variables are reported as percentages. To measure the linear association between continuous variables, Pearson’s correlation test was used if variables had a normal distribution. Otherwise, Spearman’s rank correlation test was used. Three different linear regression models were estimated for the TSH, FT3, and FT4 response variables. All the regression coefficients were adjusted for several sociodemographic, clinical, and contextual baseline patients’ characteristics. Statistical tests with *p*-values < 0.05 were considered statistically significant. All the statistical analyses were performed with R Studio Statistical software version 4.1.3. The sample size was determined using the following considerations. Spada et al. (2018) reported a mean birth weight for the Italian population of 3289 g ± 481.9 g [49]. In order to detect a 5% birth weight increase in our population (mean 3453 g), we estimated, using a power of 90% and an alpha of 0.05, a total of 68 pregnant women.

## 3. Results

During the study period, 194 pregnancies were considered for our study. Of these, 51 were excluded for maternal pathologies, 29 pregnant were excluded for fetal pathologies (fetal growth restriction, stillbirth, genetic disorders, and malformations in the fetus), and 29 had incomplete data on birth weight and thyroid hormone levels. These strict exclusion criteria were selected to reduce the population bias and analyze the low-risk pregnancies. These exclusion criteria increase our confidence in the results due to the limiting of chances of influence by possible preexisting maternal or fetal abnormalities. Following the exclusions, 98 women were included in the statistical analysis (Figure 1).

Baseline characteristics of included patients are described in Table 1.

The average age was 33.37 years (IQR = 8.67), with 66.3% identified as Caucasian. The median body mass index (BMI) was 24.98 kg/m^2^ (IQR = 5.66). The median weight gain was 13 kg, with an IQR of 7.75. Overall, 15.3% were smokers at the beginning of the pregnancy, and the average gestational age was 39.40 weeks (IQR = 2.10). Table 2 shows the linear regression model for birth weight according to TSH effect adjusted by sociodemographic and clinical factors. Thyroid levels were: TSH normal range 0.4–4.0 mIU/L, (minimum and maximum 0.03–6.9 mIU/L), FT3 normal range 2.0–4.4 pmol/L, (minimum and maximum 1.1–6.32 pmol/L), and FT4 normal range 0.8–1.8 pmol/L, (minimum and maximum 8.37–22.40 pmol/L). Of the 98 women included, 12 (12.2%) exhibited subclinical thyroid dysfunction, with TSH, FT3, or FT4 values outside the normal ranges. Specifically, eight women had subclinical hypothyroidism (elevated TSH with normal FT4) and four women had subclinical hyperthyroidism (low TSH with normal FT4). Consequently, these patients were sent for endocrinological consults and follow-up during the pregnancy. None of these patients with subclinical thyroid dysfunction was excluded, and they were considered to be within the parameters of low-risk pregnancies without overt thyroid disease.

Birth weight was not significantly associated with TSH, age, ethnicity, BMI, smoking, or weight gain. Pregnancy duration showed a significant positive association with birth weight (β = 172.027, *p*-value < 0.001). These associations agree with previous studies that considered gestational age’s role in fetal growth. Table 3 displays a negative significant correlation between FT3 and birth weight. Birth weight was found to be lower with higher FT3 in early pregnancy (beta = −118.901 95% CI: −222.942 to −14.859, *p* = 0.026, Table 3).

The analysis found no significant effect of FT4 levels on birth weight (beta = −2.388, 95% CI: −29.993 to 25.217, *p* = 0.860, Table 4).

Among the covariates examined, pregnancy weeks had a substantial effect on birth weight, with a significant positive association (*p* < 0.001, Table 3 and Table 4). Other factors, such as age, ethnicity, BMI, smoking, and weight, showed non-significant associations with birth weight. Figure 2 shows the scatterplots and Spearman correlations between birth weight and each early pregnancy’s thyroid hormone level.

The FT3 scatterplot shows a negative correlation of FT3 levels with birth weight, i.e., higher FT3 levels were associated with decreased birth weight (R = −0.22, *p* = 0.029).

## 4. Discussion

Our findings evidenced maternal thyroid hormone level’s impact on birth weight, particularly for FT3 value in early pregnancy. FT3 plays a critical role in fetal growth [6,50].

### 4.1. Thyroid Function and Pregnancy

Our findings were in disagreement with Zhang et al.’s findings, which showed an association between low FT3 levels in early pregnancy and increased risk of small for gestational age [10]. High TSH and low free thyroxin levels during pregnancy have been associated with lower birth weight and abnormal placenta [51,52]. Thyroid hormones cross the placenta mainly in their free forms, while the protein-bound forms cannot easily permeate the placental barrier [53]. The placenta expresses deiodinases like type 3 iodothyronine deiodinase (D3) that inactivate FT4 and FT3 into reverse FT3 and FT2, respectively, restricting direct maternal-to-fetal passage of active hormones [54]. Several studies have indicated that maternal FT4 contributes significantly to the fetal thyroid pool under normal conditions, in contrast to FT3 and TSH transportation [55]. Other studies have indicated that maternal FT4 is highly significant, especially during early gestation [56]. In addition, other studies explored low FT3 as a risk factor for adverse neonatal outcomes such as low birth weight [57,58]. FT3 levels impact fetal growth during the first trimester, and the biological mechanism was not completely examined [28]. A partial explanation of the FT3 effect could be given with regard to FT3’s role in metabolic enhancement and energy expenditure [59]. Indeed, thyroid hormones are critical in brain maturation and fetal development, especially in the first trimester [60]. Excessive FT3 levels can enhance metabolic rates and subsequent protein catabolism, which can result in inadequately grown fetuses. FT3 deficiency can result in failed neurodevelopment and fetal growth [61]. Our finding underlines the role of thyroid hormone levels at the beginning of pregnancy. The association of high levels of FT3 with lower birth weight represents an association that could be monitored.

### 4.2. Thyroid Dysfunction and Pregnancy

Several studies on overt and subclinical hypothyroidism have explored adverse pregnancy outcomes and altered fetal metabolic pathway [62,63]. Nishioka et al. showed a link between maternal TSH levels from the first to the third trimester and reduced birth weight: the median increase in TSH was higher in the low-birth-weight group [51]. A significant number of preterm SGA babies revealed transient hypothyroidism due to impairment of the hypothalamic–pituitary–thyroid axis [64]. Nissim Arbib et al. reported that subclinical hypothyroidism in the first trimester significantly elevated the risk of preterm delivery and very low birth weight [62]. Patients with elevated FT3 should be followed for the possible development of hyperthyroidism [65,66]. Early thyroid function should be assessed even in the absence of overt dysfunction [2]. Screening for thyroid abnormality in pregnancy could identify asymptomatic pregnancies with subclinical dysfunctions. This calls for timely interventions, including the supplementation of iodine and hormonal treatment [67]. Thyroid dysfunction detection, including subclinical hypothyroidism, hypothyroidism, and hyperthyroidism in mothers, can indicate adverse outcomes like increased risk of miscarriage, preterm birth, low birth weight, gestational hypertension, and neurodevelopmental impairment [67].

The debate on universal thyroid or high-risk-based screening of pregnancy is open [68]. High-risk screening is focused on women with prior thyroid dysfunction, autoimmune disease, or obstetric complications [69]. Universal screening includes TSH evaluation in all pregnant women during the first trimester of pregnancy. The supporters of universal screening emphasize how successful this strategy has been in offering an early diagnosis of thyroid dysfunctions and their indirect impact on adverse pregnancy outcomes [68,69]. Thyroid function is dynamic during pregnancy due to the increased need for thyroid hormones and the shift in the thresholds of TSH under the influence of hCG. Guidelines suggest a TSH cutoff of 4.0 mIU/L for normal thyroid function during the first trimester [70].

### 4.3. Strengths and Limitations

Our research has different strengths. Firstly, thyroid hormones were studied during the first trimester, covering the literature gap. Several studies have focused mainly on thyroid dysfunction during the third trimester. Second, we adopted strict inclusion and exclusion criteria to rule out the main confounding factors related to preexisting maternal conditions. This study contributes to the growing concept of subclinical endocrine factors influencing pregnancy outcomes. In addition to the endocrine factors, genetic factors, epigenetic modifications, and exposure to environmental pollutants should be considered to improve the body of knowledge on birth weight and the different impact factors, particularly in early pregnancy, when the fetal thyroid gland is not functioning and transport across the placenta to the fetal brain for such hormones depends on specific mechanisms mediated by transport proteins such as MCT8 and OATP1C1, encoded by genes with potential polymorphic variations [71]. Genetic polymorphisms in these transporters modify the efficiency of thyroid hormone transport, sometimes leading to local deficiencies in critical developmental areas of the fetal brain [71]. Genetic polymorphisms may modulate the effect of maternal thyroid hormone levels and lead to impaired neurodevelopment or lower birth weight. This highlights the importance of adequate maternal thyroid function in mitigating genetic susceptibilities during pregnancy.

There are several limitations of our study. The main one is related to the retrospective study design and the reduced number of subjects. Thyroid levels during the second and third trimesters of pregnancy were not analyzed. Moreover, minor changes in FT4 and TSH within the normal physiological range were not associated with changes in birth weight in our study, which contrasts with findings from several earlier reports suggesting a possible relationship between maternal FT4 and fetal growth [72]. However, the absence of detected associations in our study might be supported by peculiarities of the population assessed or methodological features. Further studies are needed to confirm these findings. Prospective studies are required to evaluate the long-term sequelae of maternal thyroid dysfunction, such as neurodevelopmental progress and newborn metabolic status. Moreover, thyroid function should be estimated considering genetic and environmental factors, such as genetic variants in genes implicated in the synthesis, transport, or metabolism of thyroid hormones. Traditional markers, such as TSH and FT4, provide valuable information, but may incompletely reflect the full complexity of thyroid hormone activity. Additional studies could identify novel biomarkers to provide further information on thyroid functional status and its interrelationships with other endocrine axes.

## 5. Conclusions

The results of our investigation underline the role of maternal thyroid hormone levels, more precisely free triiodothyronine, in early pregnancy regarding birth weight. High levels of FT3 appear to be negatively correlated with birth weight. Further studies that confirm and define the mechanism(s) through which mild fluctuation in FT3 determines these discrepancies in birth weight are required.

## Figures and Tables

**Figure 1 biomedicines-13-00542-f001:**
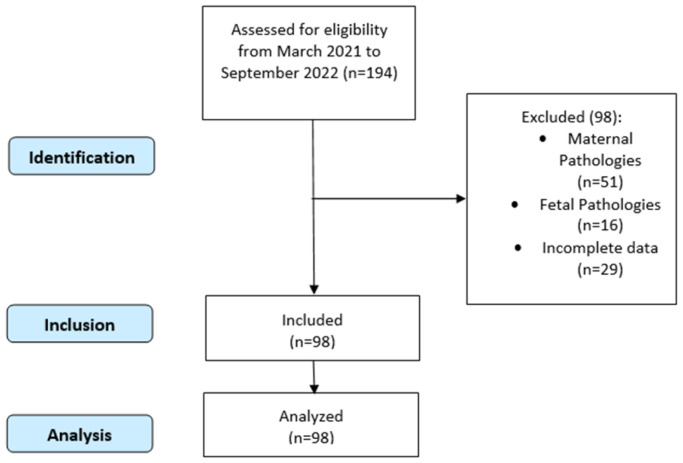
Flow diagram showing patient inclusion and exclusion in the study.

**Figure 2 biomedicines-13-00542-f002:**
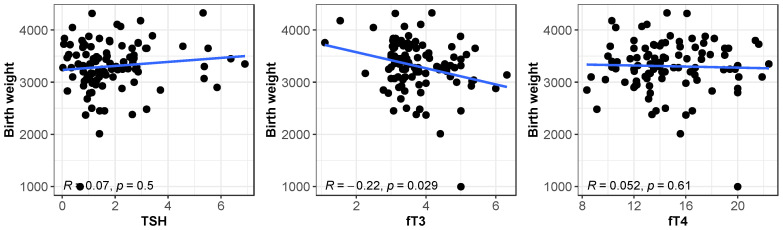
Scatterplots and Spearman correlations of thyroid function indicators on birth weight. **Left**, thyroid-stimulating hormone; **Center**, free triiodothyronine; **Right** plot: free thyroxine.

**Table 1 biomedicines-13-00542-t001:** Baseline characteristics of patients in terms of sociodemographic and clinical factors.

Characteristic	N = 98 ^1^
Age, years	33.37 (8.67)
Ethnicity, Italian	65.00 (66.3%)
Weight, kg	66.50 (16.00)
Height, cm	162.00 (7.75)
Prepregnancy BMI, kg/m^2^	24.98 (5.66)
Smoker, yes	15.00 (15.3%)
Weight gain, kg	13.00 (7.75)
Pregnancy duration, weeks	39.40 (2.10)
TSH, mIU/L	1.415 (1.685)
FT3, pmol/L	3.9 (1.375)
FT4, pmol/L	14.16 (4.07)

^1^ Median (IQR) or frequency (%).

**Table 2 biomedicines-13-00542-t002:** TSH effect on birth weight estimated by linear regression model. Estimates adjusted for sociodemographic and clinical factors.

Characteristic	Beta	95% CI ^1^	*p*-Value
TSH	28.353	−34.723, 91.428	0.370
Age	10.123	−5.640, 25.885	0.210
Ethnicity, Italian	−1.414	−189.210, 186.382	0.990
Prepregnancy BMI	16.751	−2.032, 35.533	0.080
Smoker, yes	−170.987	−423.018, 81.045	0.180
Weight gain	8.005	−2.167, 18.178	0.120
Pregnancy weeks	172.027	116.027, 228.026	<0.001

^1^ CI = confidence interval.

**Table 3 biomedicines-13-00542-t003:** FT3 effect on birth weight estimated by linear regression model. Estimates adjusted for sociodemographic and clinical factors.

Characteristic	Beta	95% CI ^1^	*p*-Value
FT3	−118.901	−222.942, −14.859	0.026
Age	9.807	−5.252, 24.866	0.200
Ethnicity, Italian	−64.931	−245.751, 115.890	0.480
BMI	13.252	−5.394, 31.898	0.160
Smoker, yes	−72.055	−332.014, 187.904	0.580
Weight increase	5.881	−4.230, 15.993	0.250
Pregnancy weeks	169.890	115.165, 224.616	<0.001

^1^ CI = confidence interval.

**Table 4 biomedicines-13-00542-t004:** FT4 effect on birth weight estimated by linear regression model. Estimates adjusted for sociodemographic and clinical factors.

Characteristic	Beta	95% CI ^1^	*p*-Value
FT4	−2.388	−29.993, 25.217	0.860
Age	8.791	−6.873, 24.455	0.270
Ethnicity, Italian	−20.089	−206.117, 165.939	0.830
BMI	17.218	−1.615, 36.051	0.073
Smoker, yes	−168.546	−421.704, 84.611	0.190
Weight gain	7.889	−2.438, 18.217	0.130
Pregnancy weeks	173.149	116.956, 229.343	<0.001

^1^ CI = confidence interval.

## Data Availability

All data generated or analyzed during this study are included in this article. Further enquiries can be directed to the corresponding author.

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
