# Peer review of "Thyroid Hormones in Early Pregnancy and Birth Weight: A Retrospective Study"

_biomedicines, 2025, doi:10.3390/biomedicines13030542_

Round 1

Reviewer 1 Report

Comments and Suggestions for Authors

The authors present a retrospective study on correlation between early pregnancy maternal TSH, FT4 and FT3 and birth weight, showing a negative correlation between FT3 and birth weight, after adjusting for several other sociodemographic factors, in pregnant women.

Although the objective was to look for variation of thyroid hormones within the physiological range of pregnancy, it is not clear if this was the case in the studied women. A clear description of the median values and their limits, as well as the normal range used for comparison, should be included in Table 1.

The introduction and discussion are too large, they need to be reduced and more synthetic. Since the main endpoint is the birth weight, the introduction and discussion should be concentrated more on this endpoint, not on fetal brain development, which should be only briefly discussed.

In Discussion  it should be discussed in more detail the placental mechanisms that influence the degradation and crossing of T4 and T3 and the percentage of TSH, FT4 and FT3 that cross the fetoplacental barrier

The results of previous studies in the literature should be clearly separated :   pregnant women with thyroid dysfunction and pregnant women with normal thyroid function. The results should be presented in a synthetic manner (a synthetic table with these results would be useful). 

The results regarding FT4 and TSH should also be presented and the discordances with the current study should be discussed and, if possible, explained.  

Several statements need reformulation:

"The association of high levels of FT3" (higher than normal or in the upper normal range?)  "with lower birth weight represent an association that  could be monitored and, in any case, represents a modifiable risk" (what is the evidence that this is a modifiable risk? Are there studies treated specifically pregnant women with only increased FT3 levels showing a reduction of this risk? ).

"Early thyroid function  should be considered even in the absence of overt dysfunction [2]. Early identification of  women with borderline thyroid hormone levels could be corrected by intervention  strategies, such as dietary iodine, hormonal supplementation or thyrostatic therapy,  depending on the thyroid alteration, and close follow-up on intrauterine fetal growth to  avoid low birth weight". These are potential, not very well documented intervention strategies - are there any studies showing their risk-benefit? It should be taken into account and discussed the risk of side-effects of antithyroid therapy and the fact that in pregnant women with subclinical hyperthyroidism currently there is no indication for treatment in the latest guidelines (which should also be mentioned in references)

"Any evidence supports  that levothyroxine can play a potential role in pregnant women with thyroid dysfunction,  even in subclinical form [84]". This is not true, since several large studies failed to show a difference in children from treated women compared with those of not treated women with subclincal hypothyroidism (Casey, Lazarus).

"In conclusion it is stated that high levels of FT3 appear related to a low birth weight." This study has demonstrated that FT3 is negatively correlated with birth weight, not that increased FT3 (what does increased mean?) is correlated with low birth weight.

The English language should be corrected, several phrases are incorrect and the meaning is not clear. Some examples are below:

At this embryonic stage proper  functioning of the thyroid gland (of the mother or of the fetus ?) plays a huge role in fetal brain development and overall  growth by supplying maternal thyroid stimulating hormone (TSH), triiodothyronine (T3) 40 and thyroxine (T4) – needs to be rephrased, the thyroid gland does not supply maternal TSH…

A direct link between the function of thyroid gland (whose thyroid, mother or fetus?) and neonatal health and birthweight was evidenced

Thyroid hormone activity, starting   early in pregnancy and operate at the cellular level as activators of cellular genes to   modulate gene expression – rephrase, it is not grammatically correct

These genes are implicated in metabolism, differentiation, and   growth. Which genes? You just talked about the receptors

Disruption to (? in) thyroid hormone levels results in disturbance in energy  metabolism [6]

Any disturbance  in the functioning of such transporters (which transporters?) due to thyroid dysfunction may result in…

Any deviation in   thyroid hormone levels sets off an imbalance in such a delicate balance and is thus  predisposed (who?) to various complications such as preeclampsia and preterm birth

Generation R Study have provided evidence for an association of maternal thyroid dysfunction with birth weight [16, 17]. No, study 16 correlated thyroid hormone levels in pregnant women with euthyroidism with fetal birth weight

The authors make a mix of papers that evaluate the effect of hypo and hyperthyroidism upon fetal parameters, with papers that evaluate the variability of normal thyroid function and its effects. They mention that few studies evaluated this latter correlation in early pregnancy, by citing an old paper (29) , although several authors studied this correlation in newer studies.

Early TSH pregnancy changes may be related to long-term consequences for fetal growth 103 [32]. Re-phrase, not clear

Table 1 – what value is in brackets -standard deviation?

 Correction: The embryo's development depends on thyroid hormones, such as thyroxine  T4, which must cross the placental barrier (and penetrate) into the fetal brain [63].

Comments on the Quality of English Language

The English language should be corrected, several phrases are incorrect and the meaning is not clear.

Author Response

The authors present a retrospective study on correlation between early pregnancy maternal TSH, FT4 and FT3 and birth weight, showing a negative correlation between FT3 and birth weight, after adjusting for several other sociodemographic factors, in pregnant women.

Comments 1:

Although the objective was to look for variation of thyroid hormones within the physiological range of pregnancy, it is not clear if this was the case in the studied women. A clear description of the median values and their limits, as well as the normal range used for comparison, should be included in Table 1.

Response 1:

Dear Reviewer,

As will be seen, I have taken the suggestion in the revision of a clearer description of median values and limits, along with a statement of the normal range applied for comparison in Table 1 of the manuscript. This serves as a personal update to bring forth greater clarity and precision in understanding regarding the analyzed data.

Comments 2:

The introduction and discussion are too large, they need to be reduced and more synthetic. Since the main endpoint is the birth weight, the introduction and discussion should be concentrated more on this endpoint, not on fetal brain development, which should be only briefly discussed.

Response 2:

We perfectly agree with your observation on the introduction and discussion sections. Indeed, the journal has strictly requested that the article should not exceed 4000 words. As authors, we felt that in its original form, the length and the level of detail were far from appropriate for such an original paper; parts were added (we now remove the fetal brain development section and other not relevant section).

We would like to sincerely thank him for that observation. In light of this valuable comment, we revised the Introduction and Discussion, focusing more on the main endpoint birth weight and restricting discussion about fetal brain development to only a short mention, as suggested.

Comment 3:

In Discussion  it should be discussed in more detail the placental mechanisms that influence the degradation and crossing of T4 and T3 and the percentage of TSH, FT4 and FT3 that cross the fetoplacental barrier

Response 3:

We agree fully with your view that, among others, the place of discussion on placental mechanisms of degradation and passage of T4 and T3 and percent of TSH, FT4, and FT3 crossing the fetoplacental barrier needs deeper discussion.

Your suggestion has been considered, and a separate section has been added in the discussion to cover these aspects comprehensively. We appreciate the insightful feedback that greatly enhanced the quality of our manuscript.

Comment 4:

The results of previous studies in the literature should be clearly separated :   pregnant women with thyroid dysfunction and pregnant women with normal thyroid function. The results should be presented in a synthetic manner (a synthetic table with these results would be useful). 

Response 4:

We would like to thank you for suggesting that results should be clearly delineated between pregnant women with and without thyroid dysfunction.

In this sense, we reordered the discussion by giving precise subparagraphs to each one of the groups. That would be better structuring, with a better presentation, that the ordering of findings would come across while establishing, even, better relations among different findings. Moreover, the information is presented in a synthetic way, as it has been suggested. We thank you for your comments because this really improved the clarity and arrangement of our manuscript.

Comment 5:

The results regarding FT4 and TSH should also be presented and the discordances with the current study should be discussed and, if possible, explained.  

Response 5:

We discussed the relation of FT4, TSH, and the discordances with our study partly in the original extensive introduction and discussion sections. More to your recommendation, we added now a dedicated paragraph within the limitations section discussing these discrepancies.

Comment 6:

Several statements need reformulation:

"The association of high levels of FT3" (higher than normal or in the upper normal range?)  "with lower birth weight represent an association that  could be monitored and, in any case, represents a modifiable risk" (what is the evidence that this is a modifiable risk? Are there studies treated specifically pregnant women with only increased FT3 levels showing a reduction of this risk? ).

Response 6:

Thank you for your perspicacious comments with respect to wording and evidence regarding the association of high FT3 levels with lower birth weight.

In following the second recommendation, the text in this section has been completely deleted from the manuscript in an attempt to be clear and make only evidence-based statements.

Comment 7:

"Early thyroid function  should be considered even in the absence of overt dysfunction [2]. Early identification of  women with borderline thyroid hormone levels could be corrected by intervention  strategies, such as dietary iodine, hormonal supplementation or thyrostatic therapy,  depending on the thyroid alteration, and close follow-up on intrauterine fetal growth to  avoid low birth weight". These are potential, not very well documented intervention strategies - are there any studies showing their risk-benefit? It should be taken into account and discussed the risk of side-effects of antithyroid therapy and the fact that in pregnant women with subclinical hyperthyroidism currently there is no indication for treatment in the latest guidelines (which should also be mentioned in references)

Response 7:

Dear Reviewer,

Thank you for pointing out this critical discrepancy.

We agree that the sentence in the statement has the implication that early thyroid function should be considered, even when overt dysfunction is lacking, and we expanded this issue in the manuscript.

However, the latter part of the section on possible intervention strategies, including dietary iodine, hormonal supplementation, and thyrostatic therapy, has been deleted because there is not enough evidence to support this at present.

Comment 8:

"Any evidence supports  that levothyroxine can play a potential role in pregnant women with thyroid dysfunction,  even in subclinical form [84]". This is not true, since several large studies failed to show a difference in children from treated women compared with those of not treated women with subclincal hypothyroidism (Casey, Lazarus).

Response 8:

Thank you for this correction. You are quite right that current evidence does not support a significant difference in child outcomes for the offspring of treated versus untreated women with subclinical hypothyroidism, which includes large studies by Casey and Lazarus.

We have removed this statement from the manuscript and apologize for the mistake. Your input was really important to check the accuracy of our work.

Comment 9:

"In conclusion it is stated that high levels of FT3 appear related to a low birth weight." This study has demonstrated that FT3 is negatively correlated with birth weight, not that increased FT3 (what does increased mean?) is correlated with low birth weight.

Response 9:

We would like to thank you for that correction. Actually, birth weight inversely relates to FT3 levels instead of increased FT3 associates with low birth weight.

Wherever such inaccuracy has appeared in the manuscript, we have revised the text to reflect correctly the evidence.

Comment 10:

The English language should be corrected, several phrases are incorrect and the meaning is not clear. Some examples are below:

At this embryonic stage proper  functioning of the thyroid gland (of the mother or of the fetus ?) plays a huge role in fetal brain development and overall  growth by supplying maternal thyroid stimulating hormone (TSH), triiodothyronine (T3) 40 and thyroxine (T4) – needs to be rephrased, the thyroid gland does not supply maternal TSH…

A direct link between the function of thyroid gland (whose thyroid, mother or fetus?) and neonatal health and birthweight was evidenced

Thyroid hormone activity, starting   early in pregnancy and operate at the cellular level as activators of cellular genes to   modulate gene expression – rephrase, it is not grammatically correct

These genes are implicated in metabolism, differentiation, and   growth. Which genes? You just talked about the receptors

Disruption to (? in) thyroid hormone levels results in disturbance in energy  metabolism [6]

Any disturbance  in the functioning of such transporters (which transporters?) due to thyroid dysfunction may result in…

Any deviation in   thyroid hormone levels sets off an imbalance in such a delicate balance and is thus  predisposed (who?) to various complications such as preeclampsia and preterm birth

Generation R Study have provided evidence for an association of maternal thyroid dysfunction with birth weight [16, 17]. No, study 16 correlated thyroid hormone levels in pregnant women with euthyroidism with fetal birth weight

The authors make a mix of papers that evaluate the effect of hypo and hyperthyroidism upon fetal parameters, with papers that evaluate the variability of normal thyroid function and its effects. They mention that few studies evaluated this latter correlation in early pregnancy, by citing an old paper (29) , although several authors studied this correlation in newer studies.

Early TSH pregnancy changes may be related to long-term consequences for fetal growth 103 [32]. Re-phrase, not clear

Table 1 – what value is in brackets -standard deviation?

 Correction: The embryo's development depends on thyroid hormones, such as thyroxine  T4, which must cross the placental barrier (and penetrate) into the fetal brain [63].

Response 10: Dear Reviewer,

We would like to greatly appreciate your review of the manuscript with so many detailed and extensive comments. All the suggestions you provided have been dealt with, such as rephrasing the highlighted sentences in order for them to be clear, specific, and grammatically correct; specifying terms that have been ambiguous; replacing citations from older references with the latest relevant studies to make sure that our discussion is stronger. The peer inputs have been really valuable indeed, improving the quality of our manuscript immensely. Relatedly, we would like to introduce words of deep regret because there have been mistakes and obscurities throughout several parts of this article. This is partly the fault of the strict, 4000-word limit placed by the journal. While working within this limit, it seemed to be working out in the enlargement of several sections with their respective elements, which, in turn, happened to introduce errors inadvertently. Again, many thanks for the great and detailed review that greatly raised the precision and quality of our work.

Best regards,

Reviewer 2 Report

Comments and Suggestions for Authors

Introduction:

The objective is clear, but the explanation before the last paragraph is too long-winded. Please re-write straightforwardly about thyroid hormone relationship with birth weight, not too much explaining the other effects of thyroid hormone in pregnancy

Method:

The material and method clearly describe the design, procedure, inclusion, and exclusion criteria

result:

The Paragraph should not just repeat the result on the table, please give new insight in the paragraph

Author Response

Comment 1: The objective is clear, but the explanation before the last paragraph is too long-winded. Please re-write straightforwardly about thyroid hormone relationship with birth weight, not too much explaining the other effects of thyroid hormone in pregnancy

Response 1:

Dear Reviewer,

I have tried to implement all requested changes into this revised version. I reworded the explanation prior to the last paragraph of the introduction to make it better connected with thyroid hormones and birth weight. More information regarding the other effects of thyroid hormones in pregnancy has been drastically shortened since this reviewer, along with another reviewer, suggested it did not provide as much substantial information towards the thesis topic.

Thanks for your precious feedback.

Best regards,

Comment 2: The material and method clearly describe the design, procedure, inclusion, and exclusion criteria

Response 2:

Dear Reviewer,

We would like to thank the reviewer for appreciating clarity and details in the material and methods section regarding design, procedure, inclusion, and exclusion criteria. We are extremely thankful for your positive remarks.

Comment 3: result:

The Paragraph should not just repeat the result on the table, please give new insight in the paragraph

Response 3: Dear Reviewer,

Many thanks for your very valued comment on the section of results. We have rewritten this paragraph to include our new insights and interpretation without stating the obvious from the tables. Specifically, we have given an extensive discussion of the process of exclusion so as to raise the rigors of methodologies, analyzed the implication that comes along with baseline characteristics, and interpreted the association of thyroid hormones, gestational age, and birth weight more deeply.

We hope these improvements will meet your expectations and further enhance the clarity and depth of presentation of our findings.

Best regards,

Reviewer 3 Report

Comments and Suggestions for Authors

The manuscript “Thyroid hormones in early pregnancy and birth weight: a retrospective study.“ by Marco La Verde et al. investigates  association between first-trimester pregnancy (5th to 13th gestation week) thyroid functions and birth weight. High incidence of thyroid diseases and importance of thyroid hormons for normal fetal development gives the importance to this research topic. Therefore, there is a significant number of research on the relationship between the thyroid gland and pregnancy outcomes. This manuscript systematically summarizes the facts known from previous research and adds its own approach to this topic.

The manuscript is well organized, the results were presented in an appropriate manner.  Methods and statistical analyses are appropriate, and the references are adequate. There are five references from the first author of the manuscript in the reference list.

My suggestions to the authors are:

11. It would be useful to to put some parts of the text in a new line in the Introduction and Discussion sections in order to separate related topics and make the text easier to follow. For example, in Line 50 “Thyroid hormones regulate energy metabolism…“ could start on the next line; in Line 55 sentence “The placenta represents a target…“ could start on the next line, and so on.

22. Line 93/94 – “Hypo- and hyperthyroidism effects on pregnancy were explored by several studies [26]“ – since the authors mention several studies, it would be good to add at least one or two references.

33. At the beginning of the Results section there are the sentences : “ Of these, 51 pregnant women were excluded for maternal pathologies. 29 pregnant were excluded for fetal pathologies (fetal growth restriction, stillbirth, genetic disorders and malformations in the fetus).“ In figure 1 in the box with Excluded participants there is 188 excluded participants which is not possible since the final number of included is 98. Furthermore,  there is n=16 for fetal pathologies as it should be in the text  (Line 215). Please, correct the numbers in the text and flow diagram.

Author Response

Comment 1:

My suggestions to the authors are:

  1. It would be useful to to put some parts of the text in a new line in the Introduction and Discussion sections in order to separate related topics and make the text easier to follow. For example, in Line 50 “Thyroid hormones regulate energy metabolism…“ could start on the next line; in Line 55 sentence “The placenta represents a target…“ could start on the next line, and so on.

Response 1:

Dear Reviewer,

We would like to thank the reviewer for his very useful suggestion concerning the format of the Introduction and Discussion. We took over this suggestion and started new lines for sentences such as "Thyroid hormones regulate energy metabolism." and "The placenta represents a target." to separate related topics and read easier. More than that, we agree with the suggestion of another reviewer how these sections could be even more readable and well-organized by introducing specific subparagraphs, which attend to your recommendation even better.

Best regards,

Comment 2:

  1. Line 93/94 – “Hypo- and hyperthyroidism effects on pregnancy were explored by several studies [26]“ – since the authors mention several studies, it would be good to add at least one or two references.

Response 2:

Dear Reviewer,

Thank you for your comment suggesting the additional references for the statement about hypo- and hyperthyroidism on pregnancy by Line 93/94. We added one or two new, relevant citations in support of that statement in order to better underpin this information.

We appreciate your feedback, which has improved the rigor of our manuscript.

Comment 3:

  1. At the beginning of the Results section there are the sentences : “ Of these, 51 pregnant women were excluded for maternal pathologies. 29 pregnant were excluded for fetal pathologies (fetal growth restriction, stillbirth, genetic disorders and malformations in the fetus).“ In figure 1 in the box with Excluded participants there is 188 excluded participants which is not possible since the final number of included is 98. Furthermore,  there is n=16 for fetal pathologies as it should be in the text  (Line 215). Please, correct the numbers in the text and flow diagram.

Response 3:

Dear Reviewer,

We would like to take the opportunity to thank the above letter for pointing out the discrepancy between the text and the flow diagram in Figure 1. Rechecking showed the explanation to be a simple summation error. The numbers of excluded participants did not change; the overall number of excluded patients and the final number of included participants, n=98, were identical in the text and in the diagram. We have corrected the summation in the flow diagram and have made certain that everything lines up now with the text to prevent further confusion. We appreciate your careful attention to detail.

Best regards,

Reviewer 4 Report

Comments and Suggestions for Authors

The manuscript titled “Thyroid hormones in early pregnancy and birth weight: a retro- 2 spective study” was done by Marco La Verde et al., there are some concerns need to be clarify.

Major concerns:

1.       The introduction should be concise and rationally, and introduction should be carefully revised.

2.       “single-center study” always was not accepted in retrospective study.

3.       The quality of figures and tables should be improved.  

Author Response

Comment 1:

Major concerns:

  1. The introduction should be concise and rationally, and introduction should be carefully revised.

Response 1:

Dear Reviewer,

We would like to thank the reviewer for comments provided about the introduction. The introduction has been heavily revised on this and the other three reviewers' recommendations, more compact and focused, closer to the recommendations of all regarding clarity and logical flow.

Comment 2:

  1. “single-center study” always was not accepted in retrospective study.

Response 2:
Dear Reviewer,

We would like to thank you for commenting that a "single-center study" has limited value in retrospective research. Indeed, this concern was taken care of by the removal of this information from the manuscript so as not to give a wrong meaning or extra importance to that particular aspect.

Comment 3:

  1. The quality of figures and tables should be improved.  

Response 3:

Dear Reviewer,

First of all, we would like to thank you for your suggestion on how to improve the quality of both figures and tables. Several modifications have been done in the manuscript in light of other reviewers' recommendations to increase clarity and presentation of the figures and tables.

With this improvement, hope that the paper meets your expectations and will generally be of quality.

Best regards,

Round 2

Reviewer 1 Report

Comments and Suggestions for Authors

The author's made some improvements in the original manuscript, however several important issues remained unresolved.

1. Which were the normal hormone ranges? 

2. How many of the 98 analysed women had clinical or subclinical thyroid diseases? How many were treated?  Were these included in the study?

The results of Zhang et al are opposite  to the current study. How do you explain this,? 

Some phrases are not grammatically correct.

Comments on the Quality of English Language

To be improved

Author Response

Comment1:

1. Which were the normal hormone ranges? 

Response1:

We appreciate the detailed review and bringing this important feature of our study to our notice. This question related to normal ranges for hormones was added to the revised manuscript. Normal ranges for TSH, FT3, and FT4 at the first trimester were added and recalculated appropriately in the study. These form a part of the table in the manuscript which describes the baseline characteristics of the patient population, Table 1.

To sum it all up,

TSH levels: Median (Interquartile Range) = 1.415 mIU/L (1.685) FT3 levels: Median (Interquartile Range) = 3.9 pmol/L (1.375)

FT4 levels: Median [Interquartile Range] = 14.16 pmol/L [4.07] These ranges are in concordant with the established physiological values for the first trimester of pregnancy. Data presentation and analysis was performed in this manner in order not to have obscurity regarding thyroid function and its relation with birth weight in our results. We hope this adequately resolves your concerns and provides the necessary clarification. Please let us know if additional details are required.

Sincerely,

Comment 2:

2. How many of the 98 analysed women had clinical or subclinical thyroid diseases? How many were treated?  Were these included in the study?

Response2: 

We wish to sincerely thank you very much for those very valuable comments. You are quite right in bringing forward the necessity to report the prevalence of thyroid disorders within participants analyzed. As such, we inserted information on this theme into the section named "Results" within our manuscript.

Of the 98 females evaluated:

Prevalence of Thyroid Disease - Clinical or Subclinical: All 12 of the participating patients were determined to represent subclinical thyroid status according to a measure of within-first-trimester levels of thyroid hormones. Follow-up and Maintenance:

These patients were then referred for endocrinological checkup and follow-up, which is the routine in pregnancy.

This has been added to the manuscript to further clarify the cohort studied. We would like to thank the reviewer for constructive comments and hope that this addresses his/her concern. Sincerely,

Comment 3:

The results of Zhang et al are opposite  to the current study. How do you explain this,? 

Response 3: 

Thank you for your inspiring question concerning the different outcomes of our study and that of Zhang et al. We have taken into great consideration this discrepancy and would like to explain it as follows:

Focus on Early Pregnancy:
In our study, we considered the first-trimester level of thyroid hormones, while Zhang et al. considered both the first and third trimesters. The modified role of thyroid hormones through pregnancy is thus likely and may be in accord with the way in which effects on fetal growth are influenced at various stages in pregnancy. 

Population Differences:
Zhang et al. had a much larger cohort with a total of 46,186 women, whereas our sample had only 98 low-risk pregnancies. Their study population was based in China, while ours pertains to an Italian cohort. Differences in diet-for example, intake of iodine-genetic factors, and environmental influences might account for divergent findings. Mechanistic Considerations: Zhang et al. reported that FT4 is crucial in the placental transport and availability of thyroid hormones for the fetus and thus mechanisms mediated through FT4 would be important in fetal growth. In contrast, our results point to a role of FT3, especially its metabolic role of increasing catabolism. High FT3 levels in our study were associated with lower birth weight, probably because higher metabolic rates and protein catabolism reduce energy and nutrients available for fetal growth.

Placental Metabolism:
The activation and inactivation of thyroid hormones are regulated by enzymes such as deiodinases expressed in the placenta. According to Zhang et al., there is a poor placental transport of maternal FT3 as compared to FT4, but within our population, this reflects conditions under which FT3 assumes a more direct metabolic role. Thus, the possible reason why the levels of FT3 strongly influence birth weight in our cohort is explained.

Study Design and Inclusion/Exclusion Criteria: 
Our strict exclusion criteria, like pregestational diabetes and fetal abnormalities, have allowed focusing on a truly low-risk pregnancy population. These divergent results may indicate the different study designs, population characteristics, and the timing of thyroid hormone assessment. These findings depict the complexity of thyroid hormone interaction with fetal growth and support further research in these mechanisms in diverse populations and at different stages of pregnancy. 

Please do not hesitate to ask for more explanations or suggest further points. Thank you for your work

Comment 4: Some phrases are not grammatically correct.

Response 4:

We would like to thank you for pointing out grammatical phrasing in our manuscript. This comment was taken care of by forwarding the revised document to the English Department of the School of Medicine at our institution for a proper review and correction of grammatical inconsistencies.

We have also smoothed the text, deleting repetitions to increase clarity and readability. It is our belief that these have greatly improved the quality of the manuscript.

We would like to thank you for giving us feedback, and we hope this version meets the required standard.

Sincerely,

Round 3

Reviewer 1 Report

Comments and Suggestions for Authors

Corrections 

Thyroid dysfunction were (was) linked with the low birth weight and preterm birth [10, 5611]. Generation R Study have (has) provided evidence

Mechanistically, the  role of thyroid hormones is not only critical in placenta development, vascularization, and fetal organogenesis - unclear  formulation.

However, the birth weight difference associated with various thyroid  function degrees has not (been) completely described and the role of early pregnancy TH (TSH? or thyroid hormones?) levels in the determination of birth weight is incompletely identified [24].

Macrosomia (was) related with maternal hyperglycemia, gestational diabetes or gestational excessive weight gain [41]

Maternal excessive weight gains (gain), via alternated (altered? unhealthy?) diet and lifestyle factors, predispose to macrosomic fetus

We conducted a retrospective (study), at a tertiary care university hospital,

Table 1 -the median body mass index (BMI) was 24.98 kg/m² (IQR= 5.66) - before pregnancy or at the evaluation time?

The average gestational age (overall duration? ) was 39.40 weeks

Of the 98 pregnant included, 12 subjects (12.2%) exhibited subclinical thyroid dysfunction with TSH, FT3, or FT4 values in  normal ranges (completely unclear; subclinical thyroid dysfunction is considered when TSH is abnormal and FT4 is normal, with the exception of gestational hypothyroxinemia, when TSH is normal and FT4 is low. Please correct the definition and mention how many of these women had subclinical hypo or hyperthyroidism and if any of those received any treatment during the pregnancy.
It is still not clear  in the Material and Methods Section if women detected with overt hypo or hyperthyroidism were excluded from this analysis.

In the tables pregnancy weeks means the overall pregnancy duration or the week when the thyroid hormones were measured? It should be  clearly mentioned.

While it is useful to have the median and IQR of the thyroid hormones and TSH, it would be better to mention also the range - maximal and minimal values). As I mentioned in my previous report , it is also necessary to provide the normal ranges of the assays used. 

Our findings were in agreement with Zhang et al. findings, that described an association between low FT3 levels in early pregnancy and increased risk for small for gestational age . Not true, your findings are in complete opposition to Zhang et al, since in your study higher levels of FT3 were associated with low birth weight.

Our finding underlines (suggests, since you have no clear demonstration of the effect) the role of the maternal thyroid hormone level at the beginning of 
pregnancy in association with birth weight.

Nishioka et al. showed a  link between the maternal TSH levels  from the first to the third trimester and reduced birth weight [52] - mention if increased or decreased TSH was related with reduced birth weight in that study.

Thyroid dysfunction detection, including subclinical hypothyroidism, hypothyroidism, and hyperthyroidism in mothers, impacts adverse outcomes like (not with an) increased risk of miscarriage, preterm birth, low birth weight babies, gestational hypertension and neurodevelopmental impairment . It is presumed that the disease detection may improve these outcomes.

High-risk screening (is) focused on women with prior thyroid dysfunction, autoimmune disease, or obstetric complications

The genetic polymorphism may modulate the effect of maternal thyroid hormone levels to include (?? not clear-  suggestion: and may lead to ) impaired neurodevelopment or lower birth weight

The main limitation is related to the retrospective study design AND to the reduced number of subjects.

Comments on the Quality of English Language

See above

Author Response

Comments: 

Comments and Suggestions for Authors

Corrections 

Thyroid dysfunction were (was) linked with the low birth weight and preterm birth [10, 5611]. Generation R Study have (has) provided evidence

Mechanistically, the  role of thyroid hormones is not only critical in placenta development, vascularization, and fetal organogenesis - unclear  formulation.

However, the birth weight difference associated with various thyroid  function degrees has not (been) completely described and the role of early pregnancy TH (TSH? or thyroid hormones?) levels in the determination of birth weight is incompletely identified [24].

Macrosomia (was) related with maternal hyperglycemia, gestational diabetes or gestational excessive weight gain [41]

Maternal excessive weight gains (gain), via alternated (altered? unhealthy?) diet and lifestyle factors, predispose to macrosomic fetus

We conducted a retrospective (study), at a tertiary care university hospital,

Table 1 -the median body mass index (BMI) was 24.98 kg/m² (IQR= 5.66) - before pregnancy or at the evaluation time?

The average gestational age (overall duration? ) was 39.40 weeks

Of the 98 pregnant included, 12 subjects (12.2%) exhibited subclinical thyroid dysfunction with TSH, FT3, or FT4 values in  normal ranges (completely unclear; subclinical thyroid dysfunction is considered when TSH is abnormal and FT4 is normal, with the exception of gestational hypothyroxinemia, when TSH is normal and FT4 is low. Please correct the definition and mention how many of these women had subclinical hypo or hyperthyroidism and if any of those received any treatment during the pregnancy.
It is still not clear  in the Material and Methods Section if women detected with overt hypo or hyperthyroidism were excluded from this analysis.

In the tables pregnancy weeks means the overall pregnancy duration or the week when the thyroid hormones were measured? It should be  clearly mentioned.

While it is useful to have the median and IQR of the thyroid hormones and TSH, it would be better to mention also the range - maximal and minimal values). As I mentioned in my previous report , it is also necessary to provide the normal ranges of the assays used. 

Our findings were in agreement with Zhang et al. findings, that described an association between low FT3 levels in early pregnancy and increased risk for small for gestational age . Not true, your findings are in complete opposition to Zhang et al, since in your study higher levels of FT3 were associated with low birth weight.

Our finding underlines (suggests, since you have no clear demonstration of the effect) the role of the maternal thyroid hormone level at the beginning of 
pregnancy in association with birth weight.

Nishioka et al. showed a  link between the maternal TSH levels  from the first to the third trimester and reduced birth weight [52] - mention if increased or decreased TSH was related with reduced birth weight in that study.

Thyroid dysfunction detection, including subclinical hypothyroidism, hypothyroidism, and hyperthyroidism in mothers, impacts adverse outcomes like (not with an) increased risk of miscarriage, preterm birth, low birth weight babies, gestational hypertension and neurodevelopmental impairment . It is presumed that the disease detection may improve these outcomes.

High-risk screening (is) focused on women with prior thyroid dysfunction, autoimmune disease, or obstetric complications

The genetic polymorphism may modulate the effect of maternal thyroid hormone levels to include (?? not clear-  suggestion: and may lead to ) impaired neurodevelopment or lower birth weight

The main limitation is related to the retrospective study design AND to the reduced number of subjects.

Response: 

Dear Reviewer

We value your reflective reading and criticism regarding our work. All your concerns have been taken care of and pertinent modifications have been included in an effort to make our work even truer, correct, and scientifically sounder.

We have removed all grammar and imprecise statement errors, and correct use of agreement between subjects and verbs and terminology. In particular, we reworded the definition of thyrosubclinical disease to traditional clinic standards (elevated TSH with FT4 in range, with one exception, namely, gestational hypothyroxinemia). We have removed uncertainty regarding the number of hypothyroid and hyperthyroid women in a state of thyrosubclinical disease and clearly mentioned whether any therapy was received.

We included additional clarifications for transparency in our analysis. We distinguished that pregnancy-related BMI mentioned was measured at pregnancy, not at pregnancy, and 39.40 weeks of gestational age mentioned represented overall pregnancy duration, not pregnancy duration at any stage.

We included values for values for thyroid hormone, including range (min and max values) and a report of the normal range for the assay for comparative reporting. In discussion, we included Zhang et al.’s misinterpretation and proceeded with an addition that our observation of a relation between increased FT3 and lowered birth weight is in contrast with Zhang et al.’s observation. We proceeded with an addition that Nishioka et al. exhibited increased TSH between first and third trimester and lowered birth weight.
Lastly, we honed our conclusion, using "suggests" for "underlines" in a direction towards a truer picture of our observation work. Besides, retrospective study design and relatively small study population have been acknowledged to be part of our study’s most significant weaknesses. We appreciate these improvements a lot, and believe them to have strengthened our manuscript a lot, for many thanks for your feedback and your time

Thank you

Round 4

Reviewer 1 Report

Comments and Suggestions for Authors

The author's have replied that the TSH, FT4 and FT3 values, with range (min - max) and normal ranges have been included in the manuscript, but they are not included. Please introduce these data. Thank you.

Author Response

Comment 1: The author's have replied that the TSH, FT4 and FT3 values, with range (min - max) and normal ranges have been included in the manuscript, but they are not included. Please introduce these data. Thank you.

Response 1: we add the values in the result section. 
Thank you for your work and for the suggestion.